# Adherence to and the Maintenance of Self-Management Behaviour in Older People with Musculoskeletal Pain—A Scoping Review and Theoretical Models

**DOI:** 10.3390/jcm10020303

**Published:** 2021-01-15

**Authors:** Anne Söderlund, Petra von Heideken Wågert

**Affiliations:** School of Health, Care and Social Welfare, Mälardalen University, 72123 Västerås, Sweden; petra.heideken.wagert@mdh.se

**Keywords:** adherence, maintenance, self-management behaviour, older adults, musculoskeletal pain, scoping review, behavioural change

## Abstract

(1) Background: Adherence to and the maintenance of treatment regimens are fundamental for pain self-management and sustainable behavioural changes. The first aim was to study older adults’ (>65 years) levels of adherence to and maintenance of musculoskeletal pain self-management programmes in randomized controlled trials. The second aim was to suggest theoretical models of adherence to and maintenance of a behaviour. (2) Methods: The study was conducted in accordance with the recommendations for a scoping review and the PRISMA-ScR checklist. Capability, motivation and opportunity were used to categorize the behavioural change components in the theoretical models. (3) Results: Among the four studies included, components targeting adherence were reported in one study, and maintenance was reported in two studies. Adherence was assessed by the treatment attendance rates, and maintenance was assessed by the follow-up data of outcome measures. For adherence, the capability components were mostly about education and the supervision, grading and mastery of exercise to increase self-efficacy. The motivation components included the readiness to change, self-monitoring and goal setting; and the opportunity components included booster sessions, feedback and social support. For maintenance, the capability components consisted of identifying high-risk situations for relapse and problem-solving skills. The motivation components included self-regulation and self-efficacy for problem solving, and the opportunity components included environmental triggers and problem solving by using social support. (4) Conclusion: There are several behavioural change components that should be used to increase older adults’ levels of adherence to and maintenance of a pain self-management behaviour.

## 1. Introduction

Adherence to treatment regimens is fundamental for pain self-management. The concept of adherence is related to compliance. Compliance typically reflects the extent that a patient actively follows orders given by health care providers. Adherence reflects a patient-centred way of communicating recommendations and letting the patient independently decide the extent to which they follow a treatment or advice [1]. In pain management, high levels of adherence to, for example, physical activity or exercise programmes have been shown to be highly correlated with positive outcomes [2,3,4]. There is no consensus for the definition of maintenance; however, for example, the maintenance of physical activity behavioural changes have been observed when positive changes in physical activity persisted months after a study [5]. Further, research on effective strategies for promoting adherence to and the maintenance of behaviours is lacking [5,6]. Most likely, patients’ beliefs, cognitions and behaviours play important roles in identifying barriers and finding strategies for successfully adhering to and maintaining a behaviour. Additionally, models of adherence to and maintenance of behavioural change components may support research aiming to identify these strategies and lead to positive outcomes of pain self-management.

In this scoping review, self-management is defined according to the definition provided by Barlow et al. [7] and refers to the ability to manage symptoms, treatment, physical and psychosocial consequences, and lifestyle changes regarding living with chronic conditions, monitoring the condition and effects on cognitive, behavioural and emotional responses. Du et al. [8] presented a systematic review and meta-analysis of self-management for chronic pain. All studies included were published between 1985 and 2009, and most of the study groups were younger than 65 years. Du et al. [8] showed that of the 19 randomised controlled trials (RCTs) included, 13 reported values of adherence to self-management programmes ranging between 56% and 97%. The attrition rates varied between 4% and 43% in all 19 studies. However, none of the studies reported adherence facilitation as a part of an intervention, and only in one study did the intervention explicitly include a maintenance component. Behaviour change techniques have been used quite extensively in the self-management programme context for patients over 18 years old who have chronic low back pain and arthritis [9], but do not explicitly support adherence to or the maintenance of pain self-management behaviours. Eisele et al. [10] studied behavioural change techniques to support adherence to physical activity in patients >18 years of age with chronic musculoskeletal conditions and found some evidence that behavioural change techniques can be used for this purpose. Pain self-management behaviours are also increasingly important for older adults, i.e., those >65 years, and have been previously studied [11,12,13]. Adhering to and maintaining pain self-management behaviours is as important for older adults as it is for other adults. Additionally, behavioural change techniques to support adherence and maintenance should be of interest to healthcare staff. However, these research questions have not been studied by summarizing the existing literature. Furthermore, research-facilitating and clinical practice-facilitating theoretical models of useful behavioural change components for supporting adherence to and the maintenance of a behaviour are non-existent. Thus, the first aim of the present scoping review was to study adherence to and the maintenance of pain self-management programmes in older adults, with study group’s mean age >65 years, with musculoskeletal pain, participating in randomized controlled trials. The second aim was to suggest theoretical models for adherence to as well as the maintenance of a behaviour, and the results of both aims can stimulate future research.

## 2. Materials & Methods

The study was conducted in accordance with the recommendations for a scoping review [14,15], and the PRISMA-ScR checklist was used to report the results [16,17]. This scoping review has been registered in the Open Science Framework, OSF registry (registration DOI 10.17605/OSF.IO/MUZF9) [18].

The methods for the first aim and second aim are presented separately.

### 2.1. Methods, First Aim

#### 2.1.1. Eligibility Criteria

The inclusion criteria for the papers were as follows:

Randomized controlled trials published in peer-reviewed journals, studies including participants with musculoskeletal pain with a study group’s mean age of >65 years, studies in which interventions regarding adherence to and the maintenance of a pain self-management programme were assessed and/or adherence to and/or the maintenance of a behaviour was measured, and studies written in English. Additional inclusion criteria were that the self-management programme aimed to improve individuals’ abilities regarding at least three of the six topics below, as defined by Barlow et al. [7], and the topics were presented as a part of a self-management programme:
manage symptoms;manage treatment;manage physical and psychosocial consequences;manage lifestyle changes regarding living with chronic conditions;monitor the condition;affect cognitive, behavioural and emotional responses.

#### 2.1.2. Information Sources and Search Strategy

To identify relevant studies, the MEDLINE, PsycINFO, and CINAHL Plus databases were searched via EBSCOhost, and the PubMed and Web of Science Core databases were searched on several occasions, with the final search for the first aim being performed on the 19 October 2020. The searches were conducted with relevant MeSH search terms. The search strategy is shown in Table 1.

#### 2.1.3. Selection of Sources of Evidence

The first author searched the databases. All eligible studies’ titles and abstracts were screened by the first author and discussed with the second author before the studies were selected for inclusion in the next step. If the decision for inclusion was uncertain, the study was included in the next step. In the second step, the full texts of the papers were downloaded to assess their relevance regarding self-management programmes. The studies’ self-management programmes were evaluated by the first author, and all decisions were made after discussion with the second author. Whether aspects of the self-management program, as defined by Barlow et al. [7], were addressed in the study was assessed, and each aspect was scored as yes, no, partly, or unclear. A minimum of three scores of yes (=half of the 6 items) were needed to include the study.

In the third step, all potential studies were assessed by both authors, and the final decision for inclusion was made in agreement.

#### 2.1.4. Data Charting Process and Parameters

The data were tabulated with relevant headings in a table (Table 2) according to the first aim of the scoping review. The first charted study also worked as a pilot test for charting. After charting this study, the last column label was clarified by adding “…results of the outcomes for adherence and maintenance”. The data were charted by the first author, and the correctness of the data was checked by the second author.

Table 2 includes information on the following variables: reference, country, aim, study population, experimental intervention including intervention for adherence and maintenance and the fulfilment of criteria for being a self-management program, control intervention, results of the self-management programme on patient outcomes, measurement method, and results of the outcomes for adherence and maintenance.

#### 2.1.5. Synthesis of Results

A critical quality appraisal was not conducted. The characteristics of all the studies included, listed in detail above in the data charting process and parameters section, are summarized in Table 2.

### 2.2. Methods, Second Aim

#### 2.2.1. Eligibility Criteria, Information Sources and Search Strategy for the Theoretical Models for Adherence to and Maintenance of Behavioural Change

First, we tried to identify systematic reviews on adherence to pain self-management in the chronic pain context at ages other than those included in our scoping review since there were no systematic reviews in the topic regarding our target group, which we had noticed when selecting studies for the first aim. The MEDLINE, PsycINFO, and CINAHL Plus databases were searched via EBSCOhost, and the PubMed and Web of Science Core databases were searched on several occasions for articles published within the last 10 years. Second, since no such reviews were found, we used systematic reviews on adherence to physical activity and/or exercise in people with chronic musculoskeletal pain and searched the PubMed database again for this purpose, with the final search being performed on the 26 October 2020. The reason that physical activity/exercise adherence was selected was that physical activity/exercise is often a component of pain self-management programmes.

The results for the search terms were as follows:exercise, adherence, pain, systematic review = 71 hits;physical activity, adherence, pain, systematic review = 76 hits;exercise, adherence, chronic pain, systematic review = 19 hits;physical activity, adherence, chronic pain, systematic review = 23 hits.

Nine relevant systematic reviews were identified from the above results [10,19,20,21,22,23,24,25,26].

We first conducted a similar search on the maintenance of pain self-management in individuals with chronic pain of ages other than those included in our scoping review’s first aim. We found one review [27] that summarized occupation and activity-based health management and maintenance. The screening of this review showed that the studies in the review did not include any isolated components (i.e., specific components with the purpose of increasing specific behaviour) to increase maintenance. The studies measured the long-term effects of different interventions and discussed the maintenance of behavioural changes according to these results, without pointing out any techniques to improve maintenance. Thus, the review was excluded. Second, we searched systematic reviews on the maintenance of physical activity and/or exercise behaviour in individuals with chronic musculoskeletal pain and found the following results with the search terms:exercise, maintenance, pain, systematic review = 8 hits;physical activity, maintenance, pain, systematic review = 13 hits;exercise, maintenance, chronic pain, systematic review = 4 hits;physical activity, maintenance, chronic pain, systematic review = 4 hits.

The screening showed that none of these reviews were relevant for our second aim.

Thus, we searched relevant reviews and books for theoretical reasoning for the maintenance of behavioural changes. Two reviews [28,29] and three books with relevant chapters [30,31,32] were included to provide insight on the modelling of the impacts of maintenance.

#### 2.2.2. Synthesis of Results

The interventions were interpreted analytically to determine whether they had isolated components for influencing adherence or maintenance of behaviours. The isolated intervention components for adherence to or the maintenance of behaviours were recorded separately, and the components were categorized according to the framework presented by Michie et al. [42] to understand behavioural changes. The three categories were capability, motivation and opportunity, all of which can influence behaviour, and vice-versa behaviour can influence all three factors. Additionally, capability and opportunity can influence motivation [42]. Capability consists of a person’s physical and psychological knowledge and skills needed for the capacity to engage in a target behaviour/activity. Motivation includes cognitive, emotional, and reflective decision making as well as habitual processes facilitating and directing behaviour/activity. Opportunity consists of behaviour prompting factors external to the person. In the final step, the categorized components were presented in two theoretical models, one for adherence to and one for the maintenance of behaviours, as inspired by Michie et al. [42].

## 3. Results

### 3.1. The First Aim

#### 3.1.1. Selection of Sources of Evidence

Five databases were searched. Figure 1 presents the PRISMA chart for the first aim’s study selection process and number of studies; see Appendix A
Table A1 for the reasons that full-text papers were excluded. In total, 4 studies were included in this scoping review to answer the research question in the first aim. Three of the four studies required additional references to determine the details of the intervention. These references for intervention details were found from the original studies’ reference lists. Thus, four studies reported in nine studies were included.

#### 3.1.2. Characteristics and Results of the Individual Sources of Evidence

The included studies were conducted in the UK [33], Canada [34,35], the USA [38,39,40,41] and Australia [36,37]. The target groups were mostly women, and long-term osteoarthritis was the main pain source. The participants’ ages varied, with the mean being between 66 and 74 years. None of the four studies fulfilled all of Barlow et al.’s [7] criteria for self-management programmes. The follow-up time for the pain self-management programmes ranged from 6 to 12 months. The magnitude of differences between the experimental and control conditions ranged from no differences to differences in some of the outcome variables.

Intervention components targeting adherence to pain self-management behaviour were reported by Nicholas et al. [36], and the maintenance of pain self-management behaviour was reported by Laforest et al. [34] and Vitiello et al. [38]. Adherence to exercise behaviour was measured in three studies [33,36,38] by the percentage attendance in the intervention. The maintenance of the effect was measured in all four studies [33,34,36,38]. No studies reported any measures for the maintenance of pain self-management behaviour. The characteristics and patient outcomes of the four randomized controlled trials included regarding adherence to and the maintenance of pain self-management programmes that met the criteria for being a self-management programme and were targeted for patients with musculoskeletal pain are presented in Table 2.

### 3.2. The Second Aim

#### 3.2.1. Selection of Sources of Evidence

Five databases were searched to carry out the second aim. Nine systematic reviews were included in developing the model for adherence to behaviour, and two theoretical papers and chapters from three books were included for the model of the maintenance of behaviour. Figure 2 presents the PRISMA chart for the second aim’s study selection process and number of studies and book chapters. Michie et al. [42] used capability, motivation and opportunity as important factors for a behaviour to categorize the collected data.

#### 3.2.2. Characteristics and Results of Individual Sources of Evidence

A total of seven components from the included systematic reviews regarding the capability of adherence to exercise behaviour were assessed (Table 3). The components that were reported mainly included education on pain and health-related aspects; supervised, individualized exercise programmes with written instructions; graded exercise and activity; and the mastery of an exercise programme to increase self-efficacy. Motivation in adhering to exercise behaviour consisted of six components (Table 4). The participants’ readiness to change, self-monitoring and goal setting were the components most often presented in the systematic reviews included. There were three opportunity-related components (Table 5), i.e., booster sessions with problem solving discussions, feedback and social support. Table 3, Table 4 and Table 5 present the reported adherence to exercise behaviour components categorized according to the behavioural wheel presented by Michie et al. [42] regarding capability, motivation and opportunity.

There were four components from the papers and book chapters included regarding the capability of maintaining a behaviour (Table 6). Primarily, components such as identifying high-risk situations for relapse, pain coping skills and skills for problem solving in high-risk situations and lapses were presented. Motivation to maintain a behaviour consisted of six components (Table 7). Self-regulation and self-efficacy for problem solving in high-risk situations to address both barriers and relapse were most often presented. There were two opportunity-related components (Table 8), i.e., environmental triggers for relapse and problem solving in high-risk situations and lapses in using social support and social reinforcement from significant others. Table 6, Table 7 and Table 8 presents the reported maintenance of behavioural change components categorized according to the behavioural wheel presented by Michie et al. [42] regarding capability, motivation and opportunity.

### 3.3. Synthesis of the Results Regarding Aims One and Two

The framework for understanding a behaviour presented by Michie et al. [42] was used to develop theoretical models for adherence to, as well as the maintenance of, pain self-management programmes. The categorized components from Table 3 and Table 4 were integrated into the Michie et al. [42] framework; see Figure 3 for the model of adherence to a behaviour and Figure 4 for the model of the maintenance of a behaviour. In both figures, the components in the capability box were suggested as needed in order to increase motivation to adhere to and maintain behaviours. The components in the opportunity box are prerequisites for increasing motivation. Furthermore, there is a reciprocal impact of capability, motivation and opportunity with adherence to and the maintenance of a behaviour.

Bearne et al. [33] did not include intervention components targeting adherence to the self-management programme. They had discussions with the participants on the importance of maintaining behaviours as a part of their daily routines. Educational discussions have not been reported as a behavioural change component regarding the maintenance of a behaviour, and not surprisingly Bearne et al. [33] did not show any significant differences in treatment outcomes between usual care and the experimental condition.

Laforest et al. [34,35] did not include intervention components targeting adherence to the self-management programme. Capability-, motivation- and opportunity-related components were included in their maintenance intervention. The authors reported significantly better results from the group participating in a self-management programme with maintenance components than from the group participating in a self-management programme without these components.

No explicitly stated adherence-increasing components were described in the studies by Nicholas et al. [36,37], but the authors reported on capability-, motivation- and opportunity-related components included in the intervention. At the one-month follow-up, the rate of adherence to exercise and pain self-management was approximately 90%. No explicitly stated intervention components were reported for increasing the maintenance of the self-management programme [36,37]. However, the one-year follow-up showed that the treatment effects were maintained for several outcomes to a greater extent in the pain self-management group than in the exercise group.

A maintenance plan as part of the intervention was mentioned, but no details were shown [38,39,40,41]. The maintenance plan could have included capability-related components. No differences between the groups in the maintenance of the treatment effect were shown. No intervention components targeting adherence to the self-management programme were reported, but the three groups all had high adherence to treatment sessions.

None of the studies reported any measures of the maintenance of pain self-management behaviours, and only the maintenance of the treatment effects was measured. The included studies showed that there are several behavioural change components that have not been, but should be, used for increasing adherence to and maintaining pain self-management behaviours in older adults.

## 4. Discussion

### 4.1. Summary of Evidence

Regarding the first aim, intervention components explicitly targeting adherence to pain self-management behaviours were not reported in any study, but the maintenance of pain self-management behaviours was reported in two studies by Laforest et al. [34] and Vitello et al. [38]. Adherence to exercise behaviour was measured in three studies [33,36,38], and the maintenance of outcome effects was measured in all four studies [33,34,36,38]. No studies reported any measures of the maintenance of pain self-management behaviours. For the second aim, the capability components for adherence were mostly about education and the supervision, grading, and mastery of exercise to increase self-efficacy. The motivation components for adherence consisted of the readiness to change, self-monitoring and goal setting. The opportunity-related components were booster sessions, feedback and social support. The capability components for maintenance consisted of identifying high-risk situations for relapse and problem-solving skills. The motivation components were about self-regulation and self-efficacy for problem solving. The opportunity-related components were environmental triggers and problem solving by using social support. The capability components are needed to increase motivation to adhere to and maintain behaviours. The opportunity components are prerequisites for motivation. The reciprocity between capability, motivation, opportunity and adherence to and the maintenance of a behaviour suggests that behaviours can influence personal skills, motivation and contextual factors.

A concept that is most likely to be important also for pain self-management is the enjoyment of the behaviour that an individual should perform. Considering the definition of Barlow et al. [7], for self-management in chronic conditions, we should consider the ability of factors, e.g., the physical and psychosocial consequences of a condition and lifestyle changes, to influence cognitive, behavioural and emotional responses; otherwise, it is hard to see how this kind of behaviour could be perceived as enjoyable. Enjoyable and pleasurable activities are those that keep an individual continuing a behaviour [43]. We might need even more shared decision making and person-centred approaches when tailoring adherence and maintenance behaviour interventions to identify the most enjoyable ways of self-managing pain so that the individual would adhere to and maintain behaviours.

The lack of studies targeting the older population (65+) was obvious during the literature search, and a large gap in knowledge was thereby identified in the present study. The number of individuals in the age group 65+ is increasing, and studies need to be conducted to investigate whether different behavioural change strategies for adherence to and maintenance of behaviour are needed in this population than in the younger/middle-aged populations previously studied. However, very few studies in younger populations [8,9] have addressed behavioural change components explicitly targeting adherence to and maintenance of a behaviour. Thus, even though the importance of adherence to a new behaviour and its maintenance over time has been recognized for many decades, e.g., [44], little has improved over the years.

The difficulty of measuring adherence to and the maintenance of behaviours is troublesome and should be addressed immediately. The rate of attendance was the measure used in the included studies. This kind of measure does not tell us much about adherence to a behaviour, which is a complex, dynamic and multidetermined structure of overt and covert behaviours [1]. The treatment attendance rate shows an aspect of observable behaviour, but covert behaviours that are cognitive, emotional and physiological [45] cannot be observed and thus need to be measured by outcomes other than the rate of attendance. Maintenance in the included studies was measured as the extent of outcome changes maintained, but the maintenance of a behaviour, e.g., in which different contexts the behaviour is observed and whether the new behaviour transfers to many contexts and situations, were not assessed. Furthermore, maintenance includes behaviours related to factors such as skills, self-regulation, self-efficacy and problem solving, as seen in our results. All these behavioural components could be measured as part of the maintenance of a behaviour. Thus, there is a great deal of room for improvement regarding the measurement of adherence to and the maintenance of a behaviour.

Murray et al. [5] studied mediators of behavioural changes in the maintenance of physical activity in a nonclinical population of young and middle-aged adults. They concluded that, in addition to individual capability-related aspects, environmental and social variables should also be considered when planning maintenance behaviour-increasing interventions. The review of Murrey et al. was not conducted in a population with pain and or on pain self-management behaviour, but the results can still be considered consistent with our results regarding the components in the model of the maintenance of a behaviour. There is unfortunately no high-quality evidence on the components included in either model. We do not know which components are more important than others and for whom. Studies on adherence to and the maintenance of pain self-management behaviour were not found in our literature search. Most likely, the components in the models need to be investigated further. Both adherence and maintenance models include many behavioural change components in different parts. There is no simple solution if we do need to apply all components to all patients in all contexts. However, individualization through the targeting and tailoring of behavioural change components should be strongly considered, which would be in line with a person-centred approach, and behavioural changes might thus be even better supported. Adherence can reinforce maintenance; thus, the overlapping of behavioural change components may be beneficial, as opposed to a situation in which the components are totally different. The systematic and conscious use of different components for adherence to and maintenance of behaviour is recommended, as increasing adherence only will not automatically lead to the maintenance of a behaviour.

The presented models are an attempt to promote adherence to and the maintenance of behaviours. We hope that this scoping review will stimulate research on these topics and lead to sustainable changes in behaviours, especially pain self-management behaviours, which are crucial, considering the increase in this the size of the population with musculoskeletal problems globally.

### 4.2. Limitations and Strengths

Scoping reviews can be performed to promote future research, as in the current case. One of the strengths of a scoping review is that ongoing research can be included in the review. However, we could not identify any ongoing studies, which is unfortunate since we only managed to find four studies to carry out our first aim. A strength is that we registered this scoping review, which demonstrates the transparency of our research for other researchers’ input. The limitations of the current scoping review include the absence of a quality assessment [14,15] and conclusions that are only descriptively summarized, which can have a negative impact on the reliability of the results. However, the aim of this review was not to specifically investigate the effects of adherence to and the maintenance of behavioural interventions, where quality assessments are very important. We wanted to exploratively study adherence to and the maintenance of pain self-management programmes in older adults with musculoskeletal pain that had been investigated in randomized controlled trials and to suggest theoretical models of adherence to, as well as the maintenance of, pain self-management programmes.

The search terms used might have biased the results. In our first attempts to search the literature, we narrowed the search significantly by using, at the same time, all the important terms in the aims. These searches yielded very few hits, and thus, we started again and used broader search terms, after which we added the important restrictions for study design and age group. These search results were the final and reported results. The screening of the search results for inclusion was discussed and decided by both authors, which may increase the reliability of the results. Additionally, if the study’s pain self-management programme did fulfil the criteria reported by Barlow [7], it was discussed by the authors to ensure it should be included in the study.

Furthermore, the data that were charted by the first author were checked by the second author. All these measures possibly increase the reliability of our results. We limited the search results to articles published in English; therefore, potentially relevant studies in other languages may have been overlooked. It was not easy to find the intervention components for adherence and the maintenance of a behaviour, as not all articles included details on the purpose of the intervention components. Thus, some components regarding both the first and second aims may have been overlooked, and the results might be biased due to this issue. However, the current paper is a scoping review that also includes suggestions for models of adherence to and the maintenance of a behaviour and is not a comprehensive systematic review.

## 5. Conclusions

In the included reviews and book chapters, there was no strong evidence of factors that promote adherence to and the maintenance of behavioural changes. Furthermore, only a few studies included a specific intervention component that affects adherence or maintenance. Thus, generally, we do not know how to increase adherence to or the maintenance of pain self-management behaviours. The proposed models with included behavioural change components for adherence and maintenance are suggested to stimulate future research on these topics. There are behavioural change components that have not, but should be, used to promote adherence to, and the maintenance of, pain self-management behaviours in older adults.

## Figures and Tables

**Figure 1 jcm-10-00303-f001:**
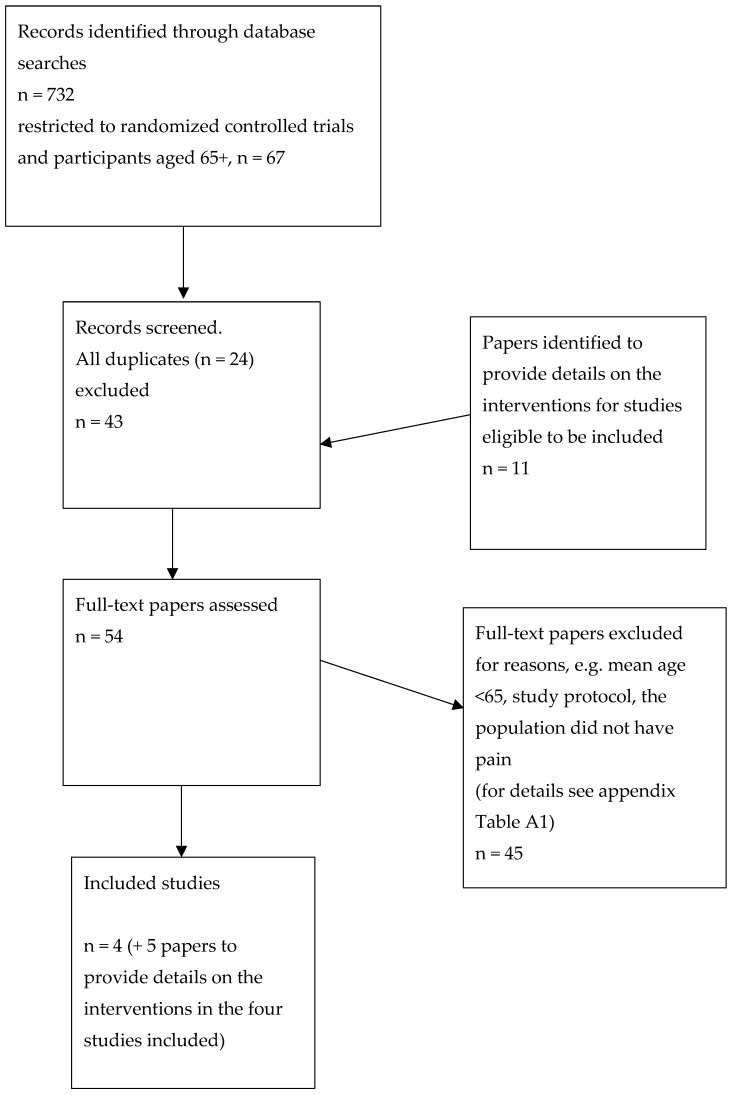
PRISMA chart of the study selection process for the first aim.

**Figure 2 jcm-10-00303-f002:**
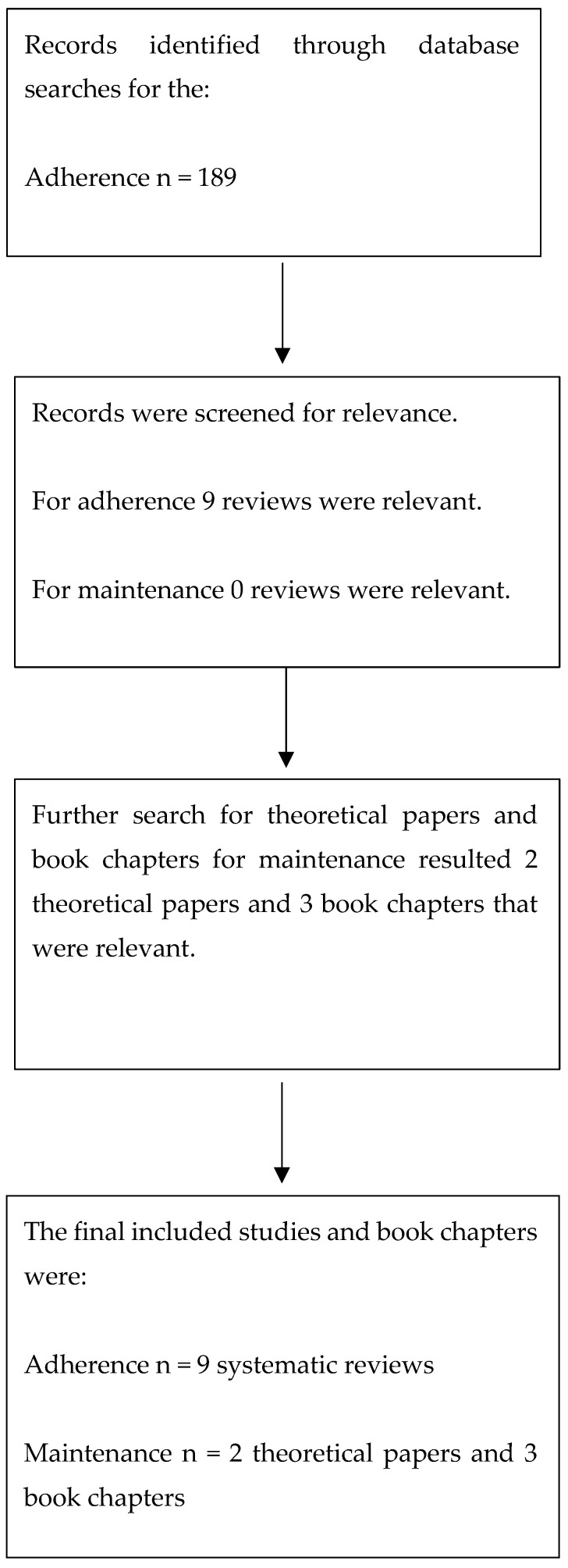
PRISMA chart of the study selection process for the second aim.

**Figure 3 jcm-10-00303-f003:**
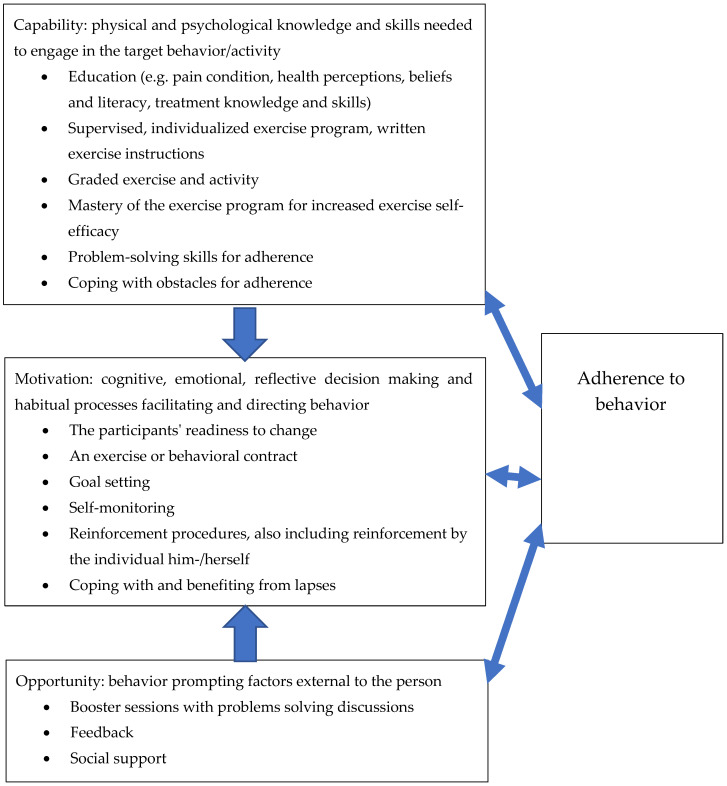
Model for adherence to a behaviour, inspired by [42]. The components were retrieved from previous studies on adherence to pain-related exercise programmes targeting participants mostly <65 years of age.

**Figure 4 jcm-10-00303-f004:**
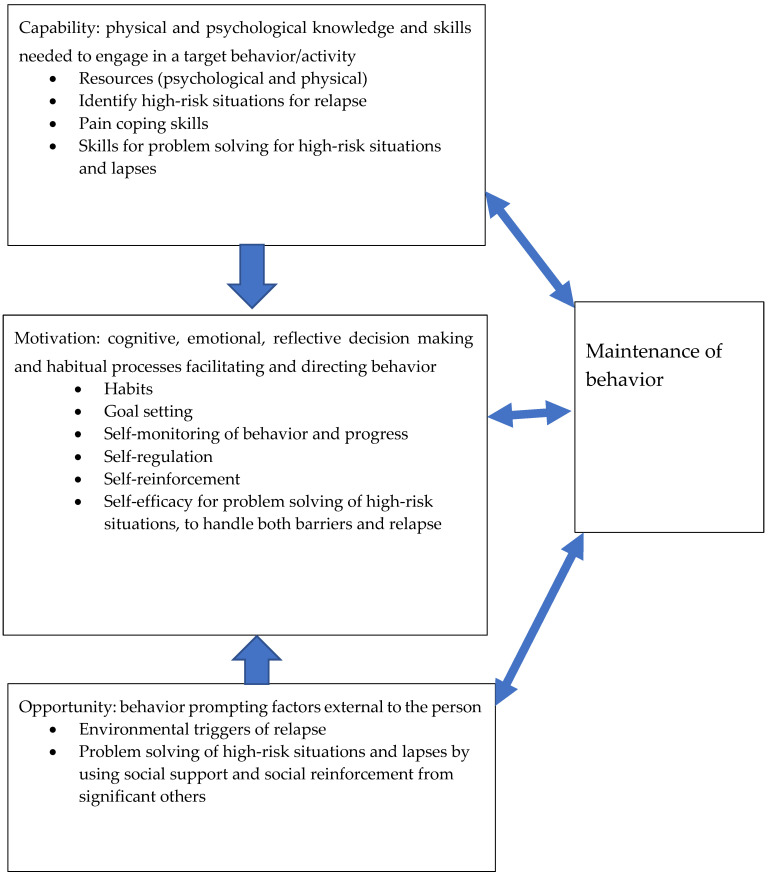
Model for the maintenance of a behaviour, inspired by [42]. The components were retrieved from previous studies on theoretical explanations of the maintenance of behavioural changes.

**Table 1 jcm-10-00303-t001:** Search strategy for adherence to and the maintenance of pain self-management programmes in older adults with musculoskeletal pain who were included in randomized controlled trials.

Databases (search date 19 October 2020) (MEDLINE, PsycINFO, CINAHL Plus) via EBSCOhost, PubMed, Web of Science Core Search restrictions: English language, “humans” years 2010–2020 Search terms EBSCOhost [All Fields]: “self-management” and pain and (adherence or maintenance) = 599 hits. “self-management” and pain and (adherence or maintenance) age limit of 65+ = 105 hits remaining from the initial 599. “self-management” and pain and (adherence or maintenance) and (RCT or randomized controlled trial) with age limit of 65+ = 34 hits remaining of the previous 105; after duplicates were removed, 27 hits remained. PubMed [All Fields]: “self-management” and pain and (adherence or maintenance) and (RCT or randomized controlled trial) = 51 hits; with age limit of 65+ = 21 hits; duplicates removed in comparison to EBSCOhost hits = 7 hits remaining. Web of Science Core [All Fields]: “self-management” and pain and (adherence or maintenance) and (RCT or randomized controlled trial) = 82 hits; + musculoskeletal = 12 hits remaining of the initial 82 hits; after the duplicates were removed in respect to the EBSCOhost and PubMed hits, 9 hits remained. *n* = the full texts of 43 studies were read and evaluated regarding the inclusion criteria (EBSCOhost; 27 hits + PubMed; 7 hits + Web of Science Core; 9 hits = 43 hits). Furthermore, 11 papers were reviewed for details on the interventions included in the studies eligible to be included in this study, yielding a total of 54 studies to be evaluated.

**Table 2 jcm-10-00303-t002:** Characteristics and patient outcomes of the four randomized controlled trials included regarding adherence to and the maintenance of pain self-management programmes in patients with musculoskeletal pain with a mean age of >65 years. Three studies required an additional two or more references to determine the details of the intervention; a total of four studies reported in nine papers were included.

Reference Country	Aim	Study Population	Experimental Intervention, Fulfilment of the Criteria for a Self-Management Programme and Intervention for Adherence and/or Maintenance	Control Intervention	Results of the Self-Management Programme on Patient Outcomes	Measurement Method and Results of the Outcomes for Adherence and Maintenance
Bearne et al. [33] UK	The aim of this feasibility randomized controlled trial was to decrease pain intensity and disability in patients with chronic hip pain.	*n* = 48 Mean age was 66 years, 34 women, mean time for hip pain was 5 years. The participants were recruited from general practitioner primary care clinics. The initial assessments were performed in physiotherapy departments.	Rehabilitation group: Usual care and 75-min group exercise that was tailored to the individual (progressively increased strengthening and stretching of the lower body, cycling, functional balance and co-ordination) and self-management sessions (education was provided through interactive discussions of coping strategies, pain control, joint protection, self-care, problem solving for lifestyle changes that would promote joint health and self-management. The importance of body weight and integrating exercises in a daily routine were emphasized. A book about the discussion topics was given) with a physiotherapist 2 times per week for 5 weeks. Outpatient treatment was provided at a primary care hospital. In the end, a home exercise programme was implemented. * Fulfilment of Barlow et al. [7] criteria for self-management program: Three scores of yes (1, 2 and 4), two scores of partly (3 and 5) one score of unclear (6). No intervention components explicitly targeted adherence to or the maintenance of the exercise or the self-management programme by methods other than discussing the importance of performing these behaviours in a daily routine.	Usual care group: Routine management by the general practitioner.	Measures: Western Ontario and McMaster universities osteoarthritis index, (WOMAC). The physical subscale of the WOMAC was the main outcome, and the general WOMAC, pain, Arthritis Self-efficacy scale, Hospital anxiety and depression scale, and Objective functional performance were assessed There were no significant differences between the groups in any of the measures. The within-group effect sizes for the experimental condition showed low-to-medium effects at the 6-week and 6-month follow ups.	Adherence measure: Adherence to the intervention programme was measured by the percentage of attendance in the experimental intervention, which was 81%. No measures of the maintenance of pain self-management behaviours were reported. The maintenance of the treatment effect showed no significant differences between groups at the 6-month follow up.
Laforest et al. [34] Detailed intervention description by Laforest et al. [35] Canada	This was a three-group randomized trial: wait list- group; self-management programme group; self-management programme with maintenance components group. The aim was to study the social reinforcement effects delivered post-intervention of a self-management programme.	*n* = 113 Mean age was 72, 90% women, participants had osteoarthritis or rheumatoid arthritis with moderate to severe pain intensity. Recruited from community health services centres by home care case managers.	There were two experimental conditions: self-management programme and self-management programme with maintenance intervention components. Self-management intervention: one hour per week for 6 weeks at the participant’s home, administered by a trained health care practitioner. Information, discussion and reinforcement was provided on exercise, relaxation, pain management, joint protection management, energy management, coping with negative emotions, available support, goal formulation, action plan, review of behavioural change success and barriers. * Fulfilment of Barlow et al. [7] criteria for self-management program: Four Yes (1, 2, 3 and 6), one Unclear (5), one No (4). No intervention components targeting adherence on the self-management programme were reported. Maintenance intervention consisted of social reinforcement with phone calls to the participants after the self-management programme. This consisted of 8 phone calls during 6 months. The calls were done by trained volunteers who had arthritis themselves. A detailed interview guide was used. The calls included discussion about action plan, revising goals, controlling pain, medication management, exercise, relaxation, positive feedback, problem-solving strategies, and energy management.	Control group: waiting list.	Outcome measures: Western Ontario and McMaster universities osteoarthritis index (WOMAC) for functional limitations. Arthritis Helplessness Inventory. Coping effectiveness. Post-intervention results showed significant differences between groups in favour to experimental condition in WOMAC and helplessness. At 10 months post randomization follow up the results were not maintained when the two experimental groups were analysed together.	No measures targeting adherence to the self-management programme were reported. No measures for the maintenance of pain self-management behaviours were reported. The maintenance of the treatment effect was measured 10 months after the randomization with the same outcome measures as mentioned for the self-management programme effect analyses. The results showed that the experimental condition with added maintenance component was significantly more effective in WOMAC compared to the condition without this component.
Nicholas et al. [36] Detailed intervention description by Nicholas et al. [37] Australia	The aim was to study the effects on disability, pain, mood, beliefs and functional reach between three groups; cognitive behavioural-based pain self-management group (PSM), exercise-attention group (EAC), and waiting list group (WL).	*n* = 141, mean age was 74 years, 63% of the participants were women, the median pain duration was 6 years, and the participants were recruited from the Pain Management and Research Centre in Sydney.	Two weekly two-hour sessions were held for four weeks. A psychologist and physiotherapist administered all the sessions together. Exercises and skills were practised during the sessions and at home. The patients received a “Manage your pain” book for information and educational purposes. The intervention included the self-monitoring of homework, the reinforcement of home tasks at each session, goal setting, activity pacing, arousal reduction, fear-avoidance management, managing flare-ups, problem solving, communication skills, stretching, aerobic, strengthening exercises, step-ups, walking. * Fulfilment of Barlow et al. [7] criteria for self-management program: Four scores of yes (1, 2, 3, 5 and 6) and one score of no (4). Adherence components were not explicitly described but the following were reported: encouragement to rehearsal of exercise and other skills at home, self-monitoring of training at home, reinforcement based on self-monitoring results. No intervention components explicitly targeted maintenance of the self-management programme.	There were two control conditions: Exercise-attention group (EAC) and waiting list group (WL) WL did not have any treatment. EAC intervention included two weekly two-hour sessions during four weeks and consisted of discussions of pain and its impact, stretching, aerobic, strengthening, step-ups, walking. A psychologist and physiotherapist administered all the sessions together. No encouragement was provided to perform the exercises at home, no self-monitoring of the exercises was performed.	Outcome measures: Roland & Morris Disability Questionnaire- Modified, Depression Anxiety Stress Scale, usual pain intensity and pain-related distress, 6-min walk test, functional reach for testing balance, catastrophizing scale of the Pain response Self-statements, Tampa Scale of Kinesio-phobia, Pain Self-Efficacy Questionnaire In short-term the PSM was significantly better in disability, pain distress, mood, pain beliefs, and functional reach compared to EAC and WL. There were no differences between EAC and WL.	Adherence measure: Attendance in the treatment sessions was set on 75% or more to have been completed the treatment. At one month follow up the percentage of non-completers varied between 11–25%, WL having the largest number of non-completers. The short-term adherence was 75% in WL, 89% in EAC and 88% in PSM. No measures for maintenance of pain self-management behaviour were reported. Maintenance of treatment effect at one year follow up: PSM group had maintained their treatment effects in pain-related disability, pain distress, pain intensity, depression and fear-avoidance beliefs significantly better than the EAC group. There was no data from the WL group at one year.
Vitello et al. [38] Detailed intervention description by Koffel et al., McCurry et al. and Von Korf et al. [39,40,41] USA	The aim was to investigate differences in pain and sleep outcomes between three groups of patients with osteoarthritis and insomnia. The three conditions were: cognitive-behavioural therapy for pain and insomnia (CBT-PI); cognitive-behavioural pain coping skills intervention (CBT-P); education-only (EOC)	*n* = 367 The mean age was 73 years, and 78.5% of the participants were women. The participants were paid volunteers who had clinically significant osteoarthritis pain (Grade II-IV in Graded Chronic Pain Scale) and insomnia and were members of a health maintenance organization “Group Health”.	The experimental groups had six weekly, 1.5 h treatment sessions for practice + homework. The CBT-PI intervention included education on pain and sleep management, sleep hygiene, sleep restriction, activity and sleep goal setting, relaxation, pleasant activity and sleep scheduling, activity pacing, a review of the schedule, problem solving, automatic thoughts, and a maintenance plan. The CBT-P intervention included education on pain and sleep management, activity goal setting, relaxation, pleasant activity scheduling, activity pacing, problem solving, automatic thoughts, and a maintenance plan. * Fulfilment of the Barlow et al. [7] criteria for a self-management program: Four scores of yes (1, 2, 3 and 6), one score of unclear (5), and one score of partly (4). No intervention components targeting adherence to the self-management program were reported. There was a “*maintenance* plan” as part of the intervention. No details were reported about this.	Six weekly, 1.5 h treatment sessions, no practices or homework. The EOC intervention included education on pain, sleep and medication management, alternative treatments, nutrition, memory and communication with health care, maintenance plan.	Outcome measures: Graded Chronic Pain Scale (pain intensity and pain interference) Insomnia Severity Index Arthritis Impact Measurement Scale Sleep efficiency with Actiwatch; % of daily time in bed. The two- and nine-month follow-ups showed that regarding some of the sleep-related measures the CBT-PI and CBT-P groups were significantly more effective than was the EOC.	Adherence measures: The attendance rate in the first session and four or more of the six possible sessions was between 92–94% in each of the three groups. Treatment acceptability level was the strongest predictor of adherence to treatment sessions. No measures for maintenance of pain self-management behaviour were reported. Maintenance of treatment effect at the nine-month follow-up: there were no group differences in pain severity or arthritis symptoms, but these issues improved in all patients over time.

* Self-management, seen as continuous and dynamic self-regulation, is defined in this scoping review by the following aspects, as reported by Barlow et al. [7]. The programmes aimed to improve individuals’ ability regarding the following aspects: 1 manage symptoms; 2 manage treatment; 3 manage physical and psychosocial consequences; 4 manage lifestyle changes regarding living with chronic conditions; 5 monitor the condition; 6 affect cognitive, behavioural and emotional responses. Regarding the fulfilment of the criteria for a self-management programme in this scoping review, at least half of the six topics above must have been addressed by the self-management programme.

**Table 3 jcm-10-00303-t003:** Adherence to exercise behaviour components categorized according to the behavioural wheel provided by Michie et al. [42] regarding capability, motivation and opportunity. Capability: Capability consists of a person’s physical and psychological knowledge and skills needed to engage in a target behaviour/activity.

Reference	Education (e.g., Pain Condition, Health Perceptions, Beliefs and Literacy, Treatment Knowledge and Skills)	Supervised, Individualised (Based on the Person’s Abilities) Exercise Programme, Written Exercise Instructions	Graded Exercise and Activity (e.g., Successively Increase the Intensity and Difficulty)	Mastery of the Exercise Programme to Increase Exercise Self-Efficacy (i.e., Mastery Increases Person’s Positive Beliefs of his/her Capability to Exercise)	Problem-Solving Skills for Adherence (e.g., Finding Solutions to Continue with a Behavior Despite Obstacles)	Coping with Obstacles for Adherence (e.g., Behavioral and Emotional Strategies to Overcome Obstacles)	Identifying Ways to Continue Exercising in the Future (e.g., Making Long-term Exercise Plans)
Beinart et al. [19]	x	x					
Eisele et al. [10]	x		x	x	x		
Ezzat et al. [20]	x	x	x	x		x	
Jordan et al. [21]	x	x	x	x			x
McLean et al. [22]	x	x		x			
Meade et al. [23]	x	x	x	x		x	
Nicolson et al. [24]	x	x	x	x		x	
Peek et al. [25]	x	x				x	
Willet et al. [26]	x		x	x	x		

**Table 4 jcm-10-00303-t004:** Adherence to exercise behaviour components categorized according to the behavioural wheel provided by Michie et al. [42] regarding capability, motivation and opportunity. Motivation: Motivation includes cognitive, emotional, and reflective decision making as well as habitual processes that facilitate and direct behaviour/activity.

Reference	The Participants’ Readiness to Change (i.e., Identify Before Start How Prepared the Person is to Change a Behaviour)	An Exercise or Behavioural Contract (e.g., Make an Agreement when to Start, and How Much the Person is Willing to Engage Her-/Himself)	Goal Setting (e.g., SMART Goals; Specific, Measurable, Achievable, Relevant, Time Bound)	Self-Monitoring (e.g., Monitor with a Diary Number of Exercise Sessions, Thoughts and Emotions before and after the Sessions)	Reinforcement Procedures, Including Reinforcement by the Individual Him-/Herself (e.g., Plan with the Person What Kind of Rewards Would Work for that Individual for Increasing Adherence)	Coping with and Benefiting from Lapses (Behavioural and Emotional Strategies to Handle Lapses and how and what to Learn from the Lapses)
Beinart et al. [19]	x				x	
Eisele et al. [10]			x	x		
Ezzat et al. [20]			x			x
Jordan et al. [21]	x	x	x	x	x	
McLean et al. [22]	x	x		x	x	x
Meade et al. [23]		x	x	x	x	x
Nicolson et al. [24]	x					
Peek et al. [25]			x	x		
Willet et al. [26]	x		x	x		

**Table 5 jcm-10-00303-t005:** Adherence to exercise behaviour components categorized according to the behavioural wheel provided by Michie et al. [42] regarding capability, motivation and opportunity. Opportunity: Opportunity consists of behaviour prompting factors external to the person.

Reference	Booster Sessions with Problems Solving Discussions (e.g., Email or Phone Based Discussions of How the Upcoming Problems Have been Solved by the Person)	Feedback (e.g., Engage a Significant Other to Give Feedback on a Performance)	Social Support (e.g., Engage a Significant Other to Follow the Person to Walking Sessions)
Beinart et al. [19]			
Eisele et al. [10]		x	x
Ezzat et al. [20]			
Jordan et al. [21]	x	x	
McLean et al. [22]			x
Meade et al. [23]	x		x
Nicolson et al. [24]	x		
Peek et al. [25]	x	x	x
Willet et al. [26]		x	x

**Table 6 jcm-10-00303-t006:** Maintenance of behavioural change components categorized according to the behavioural wheel presented by Michie et al. [42] regarding capability, motivation and opportunity. Capability: Capability consists of a person’s physical and psychological knowledge and skills needed to engage in a target behaviour/activity.

Reference	Resources (Psychological (e.g., Beliefs, Emotional Status) and Physical (e.g., Balance, Strength))	Identify High-Risk Situations for Relapse (e.g., Write down Probable Situations that Would Increase the Risk for Ending a Desired Behaviour)	Pain Coping Skills (e.g., Behavioural and Emotional Strategies to Handle the Pain Flare ups)	Skills for Problem Solving in High-Risk Situations and Lapses (e.g., what Physical and Psychological Skills are Needed for the Person to Overcome Risk Situations and Lapses
Kwasnicka et al. [28]	X			
Nigg et al. [29]	X			
Sundel et al. [30]		X	X	X
Turk [31]		X	X	X
Turk [32]		X	X	X

**Table 7 jcm-10-00303-t007:** Maintenance of behavioural change components categorized according to the behavioural wheel presented by Michie et al. [42] regarding capability, motivation and opportunity. Motivation: Motivation includes cognitive, emotional, and reflective decision making as well as habitual processes facilitating and directing behaviour/activity.

Reference	Habits (e.g., what Exercise Habits the Person has, and is There a Habit that Could be Used to Integrate Exercise with)	Goal Setting (e.g., SMART Goals; Specific, Measurable, Achievable, Relevant, Time Bound)	Self-Monitoring of Behaviour and Progress (e.g., Monitor with a Diary what Has been Done and how the Activities and Exercises are Increasing in Intensity and Difficulty)	Self-Regulation (e.g., How to Stand against Temptations to Stop the Desired Behaviour)	Self-Reinforcement (e.g., Plan with the Person what Kind of Rewards Would Work for that Individual for Increasing Maintenance)	Self-Efficacy for Problem Solving in High-Risk Situations to Address Both Barriers and Relapse (e.g., when Handling Lapses and Risk Situations Successfully Learn from Success and Reinforce One-Self)
Kwasnicka et al. [28]	X			X		
Nigg et al. [29]		X				X
Sundel et al. [30]			X	X	X	X
Turk [31]			X	X	X	X
Turk [32]			X	X	X	X

**Table 8 jcm-10-00303-t008:** Maintenance of behavioural change components categorized according to the behavioural wheel presented by Michie et al. [42] regarding capability, motivation and opportunity. Opportunity: Opportunity consists of behaviour prompting factors external to the person.

Reference	Environmental Triggers of Relapse (e.g., a TV-Program as Temptation to Skip the Exercise Session, Take Elevator instead of Stairs, Bad Weather Could Trigger Not Taking a Walk)	Problem Solving with High-Risk Situations and Lapses by Using Social Support and Social Reinforcement from Significant Others (e.g., Engage a Significant Other to Discuss What to Do with a Risk Situation and after a Lapse)
Kwasnicka et al. [28]	X	
Nigg et al. [29]	X	
Sundel et al. [30]		X
Turk [31]		X
Turk [32]		X

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
