# Peer review of "Adherence to and the Maintenance of Self-Management Behaviour in Older People with Musculoskeletal Pain—A Scoping Review and Theoretical Models"

_jcm, 2021, doi:10.3390/jcm10020303_

Round 1
Reviewer 1 Report
This scoping review explored factors that promote adherence to and maintenance of pain self-management behaviours in adults >65years. A systematic review identified 4 randomised controlled trials and identified levels of adherence and maintenance. The authors aimed to identify independent factors for adherence and maintenance.
A second aim used a broader search strategy to identify theoretical models of adherence to and maintenance of behaviours. The authors synthesised the evidence with published theoretical models of adherence and maintenance (Mitchie et al). Capability, motivation and opportunity were used to categorize the behavioural change components in the theoretical models.
The review highlights the lack of dedicated research on the topic and complexity of topic. Reviews such as the current one are important to help direct future research in the area and I commend the authors on a comprehensive review. The following suggestions are mainly directed to improving the readability of the information in this manuscript.
It feels like the aims might be better swapped as aim #2 is broader in focus and involves more than one aim. For example, the authors first reviewed a broader literature including all adults, not just those over 65years. They then used the theoretical models to categorise the components used in the RCTs. As the order may be not easily changed, can the authors add some figures to illustrate aim 2 flow.
The authors may need to introduce the theoretical models earlier eg perhaps fig 2&3 can be earlier. This will help the reader understand the specific components that were considered and perhaps how many studies were identified that considered each components. It would be also helpful for the authors to provide some examples of the components in the text. For example, how is ‘environmental triggers of relapse’ an opportunity.
The authors use the term “isololated components”. Can they clarify what is meant by isolated.
Can the authors please expand on the justification for identifying papers in all adults in the methods and comment upon any differences between the two cohorts.
The authors discuss that rate of attendance was used as the measure in the included studies. Did the authors intend to measure other aspects of adherence, such as adherence to exercises or physical activity or behaviour modifications etc?
Reference to table 2 is recommended to be in the results rather than methods sections.
Typo in Introduction – remove the word ‘a’ from “maintenance of a behaviours”
Table 2 difficult to follow lines across- suggest change in formatting.
Author Response
Reviewer 1:
Comments and Suggestions for Authors
This scoping review explored factors that promote adherence to and maintenance of pain self-management behaviours in adults >65years. A systematic review identified 4 randomised controlled trials and identified levels of adherence and maintenance. The authors aimed to identify independent factors for adherence and maintenance.
A second aim used a broader search strategy to identify theoretical models of adherence to and maintenance of behaviours. The authors synthesised the evidence with published theoretical models of adherence and maintenance (Mitchie et al). Capability, motivation and opportunity were used to categorize the behavioural change components in the theoretical models.
The review highlights the lack of dedicated research on the topic and complexity of topic. Reviews such as the current one are important to help direct future research in the area and I commend the authors on a comprehensive review. The following suggestions are mainly directed to improving the readability of the information in this manuscript.
Thank you!
It feels like the aims might be better swapped as aim #2 is broader in focus and involves more than one aim. For example, the authors first reviewed a broader literature including all adults, not just those over 65years. They then used the theoretical models to categorise the components used in the RCTs. As the order may be not easily changed, can the authors add some figures to illustrate aim 2 flow.
Thank you for your suggestion. Our purpose was to highlight the adherence and maintenance of self-management behavior in pain context for older adults and thus we wanted to have the aim targeting this as the first and most important. If there had been reviews in the topics for the second aim that were done with older adults, we would have included them for the model development. However, we had to choose “the next best thing”, i.e. studies with other than older adults and also theoretical papers and book chapters since we could not find the “best thing”. Your suggestion of the flow figure for aim 2 has been added in the results; 3.2.1. And the following sentence on lines 244-245. “Figure 2 presents the PRISMA chart for the second aim’s study selection process and number of studies and book chapters.” We hope this will be clarifying.
The authors may need to introduce the theoretical models earlier eg perhaps fig 2&3 can be earlier. This will help the reader understand the specific components that were considered and perhaps how many studies were identified that considered each components. It would be also helpful for the authors to provide some examples of the components in the text. For example, how is ‘environmental triggers of relapse’ an opportunity.
It is a tricky business to find the most perfect placing for the model figures. The two theoretical models are results belonging to the aim 2 and thus in our opinion they should emerge in the results. Our reasoning was that the models would be the results from the 3.3. Synthesis of the results regarding aims 1 and 2 and thus placed after that part of the results. But if you have an idea of where in the results the model figures would be better suited, please let us know and we change this.
Regarding the examples of the meaning of the components, we have added these in the tables 3 a,b,c and 4 a, b,c and we hope that the examples are mirroring the underlying concepts of capability, motivation and opportunity. The concept opportunity can be seen from positive and negative sides. E.g. there is an opportunity of seeing “bad weather as environmental trigger of relapse” i.e. being a reason for not take a walk.
The authors use the term “isolated components”. Can they clarify what is meant by isolated.
The following (here in red) has been added on lines 169-170 in the following sentence “The screening of this review showed that the studies in the review did not include any isolated components (i.e. specific components in purpose for increasing specific behaviour) to increase maintenance.” We have also changed the word isolated to specific in Conclusions on line 463.
Can the authors please expand on the justification for identifying papers in all adults in the methods and comment upon any differences between the two cohorts.
We believe that the reviewer is referring to the second aim methods. On lines 148-150, the sentence has been clarified with the red text: “First, we tried to identify systematic reviews on adherence to pain self-management in the chronic pain context at ages other than those included in our scoping review since there were no systematic reviews in the topic regarding our target group, which we had noticed when selecting studies for the first aim.” Thus, there were not two cohorts, i.e. we found no systematic reviews on adherence to pain self-management of older adults in the chronic pain context. These missing reviews is of course also the reason why we conducted the current scoping review.
The authors discuss that rate of attendance was used as the measure in the included studies. Did the authors intend to measure other aspects of adherence, such as adherence to exercises or physical activity or behaviour modifications etc?
We collected and reported every measure for the adherence and maintenance from the included studies. There were, sadly so, no other measures for the adherence or maintenance than the reported in table 2.
Reference to table 2 is recommended to be in the results rather than methods sections.
We agree, this was the case in the submitted manuscript, but the typesetting at the Journal have changed the placement of table 2.
Typo in Introduction – remove the word ‘a’ from “maintenance of a behaviours”
This has been corrected.
Table 2 difficult to follow lines across- suggest change in formatting.
Tables have been formatted in accordance with the requirements of this journal. Unfortunately, we probably cannot do anything about this. In the original submitted manuscript each study in table 2 started with a new page which made it easier to follow the respective study’s data sideways. I hope that the Journal would change this back to that if the paper is accepted.
Reviewer 2 Report
Comments to the author
Thanks for inviting me to review this interesting manuscript titled: “Adherence to and the maintenance of self-management behaviour in older people with musculoskeletal pain – a scoping review and theoretical models.” This research conducted a scoping review to address two main questions: 1) to study older adults levels of adherence to and maintenance of musculoskeletal pain self-management programmes in RCTs 2) to describe and synthesize theoretical models of adherence to and the maintenance of a behaviour.
The concept adherence to and the maintenance of self-management behaviour is really important not alone in the field of musculoskeletal pain, but in many other fields in healthcare. In general, less adherence to and the maintenance of self-management behaviour meaning lower quality-adjusted life years and higher healthcare costs. To address this topic a Scoping Review fits, because in chronic pain treatment adherence to and the maintenance of self-management is quite challenging and complex, and the increasing volume of studies.
However, when considering this manuscript I would address some concerns and questions.
- “A scoping review or scoping study is a form of knowledge synthesis that addresses an exploratory research one of the aspects question aimed at mapping key concepts, types of evidence, and gaps in research related to a defined area or field by systematically searching, selecting, and synthesizing existing knowledge [Colquhoun et al., 2014 Journal of Clinical Epidemiology].” Regarding the first aim of the study, the authors only include RCTs. I do not understand the rationale to only include RCT studies, especially when conducting a Scoping Review. The study selection of a Scoping Review is not linear, but rather an iterative process that involves searching the literature, refining the search strategy, and reviewing articles for study inclusion (Colquhoun 2014), therefore the question arises of the research search/aim is not too strict for a Scoping Review.
- The authors stated “Adhering to and maintaining pain self-management behaviours is as im-important for older adults as it is for other adults.” Regarding the first aim, the authors included older people (>65 years). I do not understand the rationale for this inclusion criteria. On which biological/theoretical explanation is this inclusion criteria rationale based? Other possibility is that based on pragmatic judgment, however, this scoping review only included 4 studies for the first aim of the study.
- The search of the second aim is really strict. I am wondering of the theories such as therapeutic (working) alliance, health literacy and person-centred care could help to develop a theoretical model of adherence to ass well ass the maintenance of a behaviour? Moreover, I am wondering of a more qualitative approach rather than a quantitative (content analysis) approach is more suitable for more understanding of this topic. Namely, “contextual or process-oriented data may require a qualitative content analysis approach” (Colquhoun 2014).
In summary, I think this manuscript would benefit of a more broader scope of this topic. This will increase the impact of the manuscript.
Some minor issues
- For the reader it is not clear of the inclusion criteria aim 1 is mean age of >65 or only adults > 65. Because in the manuscript both descriptions is given.
- Does the authors pilot the charting form to determine whether their approach to data extraction is consistent with the research question and purpose?
- Figure 1. Missing 5 papers in the flow-chart: Full-text papers assessed n=54| Full-text papers excluded N=45 | Included 4 studies. The +5 papers was an additional search or belonging in the current search?
Author Response
Reviewer 2:
Comments and Suggestions for Authors
Comments to the author
Thanks for inviting me to review this interesting manuscript titled: “Adherence to and the maintenance of self-management behaviour in older people with musculoskeletal pain – a scoping review and theoretical models.” This research conducted a scoping review to address two main questions: 1) to study older adults levels of adherence to and maintenance of musculoskeletal pain self-management programmes in RCTs 2) to describe and synthesize theoretical models of adherence to and the maintenance of a behaviour.
The concept adherence to and the maintenance of self-management behaviour is really important not alone in the field of musculoskeletal pain, but in many other fields in healthcare. In general, less adherence to and the maintenance of self-management behaviour meaning lower quality-adjusted life years and higher healthcare costs. To address this topic a Scoping Review fits, because in chronic pain treatment adherence to and the maintenance of self-management is quite challenging and complex, and the increasing volume of studies.
Thank you!
However, when considering this manuscript I would address some concerns and questions.
- “A scoping review or scoping study is a form of knowledge synthesis that addresses an exploratory research one of the aspects question aimed at mapping key concepts, types of evidence, and gaps in research related to a defined area or field by systematically searching, selecting, and synthesizing existing knowledge [Colquhoun et al., 2014 Journal of Clinical Epidemiology].” Regarding the first aim of the study, the authors only include RCTs. I do not understand the rationale to only include RCT studies, especially when conducting a Scoping Review. The study selection of a Scoping Review is not linear, but rather an iterative process that involves searching the literature, refining the search strategy, and reviewing articles for study inclusion (Colquhoun 2014), therefore the question arises of the research search/aim is not too strict for a Scoping Review.
Thank you for your comment. We of course agree with you of the process for a scoping review. But we wanted to highlight the gaps in research regarding the studies investigating the effects of pain self-management and also adherence and maintenance of this kind of behavior change in older adults. In our opinion, when studying adherence and maintenance of any behavior there need to be an approach for a behaviour change before one talk about adherence to and maintenance of a behaviour. These are the reasons for choosing only the RCT studies and synthesizing the existing knowledge from these.
We also, in our opinion had: “an iterative process that involves searching the literature, refining the search strategy, and reviewing articles for study inclusion (Colquhoun 2014)” that we have tried to describe in the method section with the complexity as it was.
- The authors stated “Adhering to and maintaining pain self-management behaviours is as important for older adults as it is for other adults.” Regarding the first aim, the authors included older people (>65 years). I do not understand the rationale for this inclusion criteria. On which biological/theoretical explanation is this inclusion criteria rationale based? Other possibility is that based on pragmatic judgment, however, this scoping review only included 4 studies for the first aim of the study.
It is difficult to decide where the age-cutoff for being an older adult is. We landed on >65 years which is the most general legal retirement age EU member States including UK. Many studies in our topic have been done on people that are younger than 65 years old but only a few on people older than 65 years. Our decision was also supported by the pain focus of this paper, where it is known that pain-related problems and management are not equal or alike between those younger and older than 65 years. For example, people in working ages are offered multimodal rehabilitation more often than those in retirement (ages over 65 years). We are not sure if we understood the reviewer correctly but let us know if our response is satisfactory to this comment.
- The search of the second aim is really strict. I am wondering of the theories such as therapeutic (working) alliance, health literacy and person-centred care could help to develop a theoretical model of adherence to ass well ass the maintenance of a behaviour? Moreover, I am wondering of a more qualitative approach rather than a quantitative (content analysis) approach is more suitable for more understanding of this topic. Namely, “contextual or process-oriented data may require a qualitative content analysis approach” (Colquhoun 2014).
We agree that a qualitative approach might have given us deeper understanding of model development as well as the theories raised by the reviewer. However, our ambition was not to deliver a full scale adherence and maintenance model development, which would need much greater effort and would be a paper or two of its own. We wanted to highlight that there are number of behavior change components and strategies that are important for these two concepts and that the components can be seen on different areas. We also wanted to facilitate the future research in adherence and maintenance model development which certainly could be done with a qualitative approach.
In summary, I think this manuscript would benefit of a more broader scope of this topic. This will increase the impact of the manuscript.
Some minor issues
- For the reader it is not clear of the inclusion criteria aim 1 is mean age of >65 or only adults > 65. Because in the manuscript both descriptions is given.
This has been clarified by adding “study group’s mean age >65 years” on lines: 78 and 92.
- Does the authors pilot the charting form to determine whether their approach to data extraction is consistent with the research question and purpose?
The first charted study also worked as a pilot test for charting presented in table 2. After charting this study one column label was clarified by adding “….results of the outcomes for adherence and maintenance”. We have added this in the method part in lines 131-133.
- Figure 1. Missing 5 papers in the flow-chart: Full-text papers assessed n=54| Full-text papers excluded N=45 | Included 4 studies. The +5 papers was an additional search or belonging in the current search?
Sorry about this missing information in the text. This was only present in the table 2 text. We have now added the following on lines 214-217: “Three of the four studies required additional references to determine the details of the intervention. These references for intervention details were found from the original studies’ reference lists. Thus, four studies reported in nine studies were included.” We hope this would be clarifying.
Round 2
Reviewer 2 Report
Thank you for addressing carefully my queries and comments. Although I am still convinced that this scoping review with a qualitative analysis, broader study selection, and a less restriction of age will increase the potential impact of the paper, beside that, I can also agree with the authors arguments. And that this scoping review may facilitate the future research in adherence and maintenance model development which certainly could be done with a qualitative approach.